# *SMARCA4* Depletion Induces Cisplatin Resistance by Activating YAP1-Mediated Epithelial-to-Mesenchymal Transition in Triple-Negative Breast Cancer

**DOI:** 10.3390/cancers13215474

**Published:** 2021-10-30

**Authors:** Jihyun Kim, Gyubeom Jang, Sung Hoon Sim, In Hae Park, Kyungtae Kim, Charny Park

**Affiliations:** 1Research Institute, National Cancer Center, Ilsanro, Ilsandonggu, Goyangsi 10408, Korea; blrusti@ncc.re.kr (J.K.); jkb0329@ncc.re.kr (G.J.); simsh@ncc.re.kr (S.H.S.); parkih@korea.ac.kr (I.H.P.); 2Department of Hemato-Oncology, Division of Internal Medicine, Korea University College of Medicine, Guro Hospital, Seoul 08308, Korea

**Keywords:** triple-negative breast cancer, cisplatin resistance, molecular subtype, *SMARCA4*, epithelial-to-mesenchymal, Hippo-YAP/TAZ pathway

## Abstract

**Simple Summary:**

*SMARCA4* mutations were over-representative in cisplatin resistance and metastatic triple-negative breast cancer (TNBC). Additionally, *SMARCA4* inactivation induced the mesenchymal-like subtype TNBC. The epithelial-to-mesenchymal transition and Hippo-YAP/TAZ pathways were activated in *SMARCA4* inactivation samples of both *SMARCA4* knockout cell lines and TNBC patients. In *SMARCA4* knockout cells, the YAP1 inhibitor verteporfin suppressed *YAP1* target genes. This study depicts the clinical importance of *SMARCA4* depletion in TNBC and suggests YAP/TAZ as a novel target for cisplatin-resistant patients.

**Abstract:**

The role of *SMARCA4*, an ATPase subunit of the SWI/SNF chromatin remodeling complex, in genomic organization is well studied in various cancer types. However, its oncogenic role and therapeutic implications are relatively unknown in triple-negative breast cancer (TNBC). We investigated the clinical implication and downstream regulation induced by *SMARCA4* inactivation using large-scale genome and transcriptome profiles. Additionally, *SMARCA4* was knocked out in MDA-MB-468 and MDA-MB-231 using CRISPR/Cas9 to identify gene regulation and a targetable pathway. First, we observed an increase in *SMARCA4* mutations in cisplatin resistance and metastasis in TNBC patients. Its inactivation was associated with the mesenchymal-like (MSL) subtype. Gene expression analysis showed that the epithelial-to-mesenchymal transition (EMT) pathway was activated in *SMARCA4*-deficient patients. Next, the Hippo pathway was activated in the *SMARCA4* inactivation group, as evidenced by the higher *CTNNB1*, *TGF-β*, and *YAP1* oncogene signature scores. In *SMARCA4* knockout cells, EMT was upregulated, and the cell line transcriptome changed from the SL to the MSL subtype. *SMARCA4* knockout cells showed cisplatin resistance and Hippo-YAP/TAZ target gene activation. The YAP1 inhibitor verteporfin suppressed the expression of *YAP1* target genes, and decreased cell viability and invasiveness on *SMARCA4* knockout cells. *SMARCA4* inactivation in TNBC endowed the resistance to cisplatin via EMT activation. The YAP1 inhibitor could become a novel strategy for patients with *SMARCA4*-inactivated TNBC.

## 1. Introduction

Triple-negative breast cancer (TNBC) is known to have an aggressive nature with a higher recurrence rate, and more heterogeneous genomic characteristics than other breast cancer subtypes [1,2]. Over 30% of TNBC patients present metastases during the disease course or become refractory to treatment [3,4]. Unfortunately, there is no effective target therapy for metastatic TNBC [4]. Moreover, 90% of therapy including cisplatin failed in TNBC metastasis, so chemotherapy resistance is a significant hurdle for metastatic TNBC treatment [5]. Although novel and combinatorial immunotherapies have been recently proposed, the platinum-based drug cisplatin remains a first-line treatment for TNBC patients [3]. After metastasis, the mutations presented after chemotherapy are not markedly different compared with primary TNBC [6]. These characteristics increase the difficulty of treating cisplatin-relapsed TNBC.

TNBC is the most heterogeneous type among breast cancers. TNBC has been classified into several molecular subtypes based on the gene expression profile [7]. Various TNBC subtypes have been proposed with distinct characteristics [7,8,9], but there are at least four main subtypes, which are as follows: mesenchymal-like (MSL), which shows EMT; basal/stem-like (SL), which exhibits stemness and rapid cell cycle progression; LAR is AR positive; and immune activation (IM) shows immune cell infiltration [7,8,9,10]. The IM and LAR types were clearly considerable for immune checkpoint inhibitor or AR-targeted treatment, respectively. The global pathway regulation or drug sensitivity of the subtypes were investigated, and it was revealed that platinum-based treatment, such as cisplatin, was sensitive in the SL type [8,9,10]. In contrast, MSL exhibited resistance in cisplatin. However, it is unclear which upstream regulators are involved in cisplatin resistance and what the next therapy for overcoming resistance could be.

The epigenetic regulator *SMARCA4* is a member of the SWI/SNF chromatin remodeling family that has been implicated in loss-of-function (LoF) mutations or downregulation in various cancers [11,12,13]. Similarly, biallelic *SMARCA4* mutations are drivers of ovarian carcinoma [12]. Fusions and deletions are detected in approximately 5~10% of lung cancer [14]. In non-squamous lung cancer, *SMARCA4* LoF is associated with worse overall survival (OS) and increased sensitivity to platinum-based chemotherapy [15,16]. It is clear that most *SMARCA4* mutations are involved in LoF to decrease expression [15,16]. However, the downstream regulation according to *SMARCA4* mutation is unclear and controversial in various cancers [13]. A TCGA pan-cancer meta-data analysis suggested that *SMARCA4* does not consistently function as a tumor suppressor across various cancer types [17]. Additionally, another previous study revealed roles for *SMARCA4* in telomere organization, binding topologically associating domain (TAD) boundaries, and regulating extracellular matrix (ECM) genes from the human mammalian epithelial cell line MCF-10A [18]. However, clinical and therapeutic studies of *SMARCA4* are still lacking for TNBC.

Here, we investigated the clinical relevance of *SMARCA4* mutations in our cohort (*n* = 37) who received cisplatin therapy, and an additional large-scale cohort of TNBC patients collected from multiple cohorts (*n* = 3455) [19,20,21,22]. More recurrent variants were interrogated in cisplatin resistance and metastatic TNBC patients. Next, regulated pathways according to *SMARCA4* activities were explored from a large-scale gene expression profile (*n* = 957) [7]. Additionally, we identified that *SMARCA4* activities determine the TNBC molecular subtype. To evaluate previously identified downstream pathways, we generated *SMARCA4* knockout (KO) TNBC cell lines using the CRISPR/Cas9 system, and analyzed gene expression from RNA-seq. Finally, we explored targetable pathways that were sensitive to *SMARCA4* depletion. Our investigation delineates that *SMARCA4* inactivation induces a specific TNBC molecular subtype and is associated with cisplatin resistance. Moreover, we suggested a sensitive drug for cisplatin-resistant TNBC to inhibit the downstream pathway regulated by *SMARCA4* KO.

## 2. Materials and Methods

### 2.1. TNBC Patient Cohort and Sample Preprocessing

Tumor and matched blood samples (*n* = 37) were obtained from advanced refractory TNBC patient cohort. All patients had undergone cisplatin chemotherapies in the metastatic setting (Appendix A). DNA was isolated from tumor tissues and paired blood samples using AllPrep DNA Mini Kit and QUAamp DNA Blood Mini Kit according to manufacturer’s protocol (Qiagen, Valencia, CA, USA). Sequencing libraries were generated using the SureSelectXT Library Preparation Kit (Illumina, San Diego, CA, USA) for analysis on the HiSeq2500 system (Illumina, San Diego, CA, USA). Cisplatin responder was defined as a case in which the response was stable disease or higher (complete response or partial response) and lasted for more than 6 months. This study was approved by the National Cancer Center Institutional Review Board (IRB No. NCC2016-0221).

### 2.2. Variant Analysis of Cisplatin-Treated TNBC Patients

To identify genomic variants, whole-exome sequencing (WES) of 37 TNBC patients was performed. FASTAQ files were aligned to the human reference genome hg19 using BWA after low-quality trimming using Trimmomatic [23,24]. Alignments were marked for duplicates, recalibrated, and realigned using PICARD [25]. Next, we executed Mutect2 and Oncotator to call and annotate somatic mutations [25,26]. Copy number variant (CNV) analysis was performed by EXCAVATOR2 version 1.1.2, and the peak regions were identified using GISTIC2 [27,28]. Variant enrichment according to cisplatin response was evaluated using the chi-square test. In addition to our cohort, we obtained the frequency of mutated genes in primary and metastatic breast cancer from public datasets, including TCGA (*n* = 1084), the MSK cohort (*n* = 1918), the INSERM cohort (*n* = 216), and the MBC project (*n* = 237) [19,20,21,22]. Finally, mutation profile was acquired from 2002 primary and 1453 metastatic TNBC patients. We interrogated mutation genes observed significantly more in metastasis by chi-squire test. For survival analysis, patients were divided by cisplatin response, and survival curves were fitted using the Cox regression model in the R package ‘survival’.

### 2.3. Gene Expression Analysis Based on SMARCA4 Activity and BC Subtype

To ascertain the downstream effects according to *SMARCA4* inactivation, we analyzed gene expression profiles from a TNBC patient cohort (*n* = 957) inclusive of the four subtypes (MSL: luminal androgen receptor (AR)-positive; LAR: immunomodulatory; IM; SL) [7]. First, we divided samples by *SMARCA4* mRNA expression into inactivation and activation groups. Next, we confirmed that two *SMARCA4* groups show the enrichment of subtypes using Fisher’s exact test. Pathway activity scores for each sample were estimated by gene set variation analysis (GSVA) based on the MSigDB hallmark gene set collection including EMT, stemness, and cisplatin resistance [7,29,30,31]. We classified patients into inactivation and activation groups by *SMARCA4* gene expression upper (log-scale expression > 9.71, *n* = 244) and lower (log-scale expression < 9.08, *n* = 237) tertile. Then, we analyzed the pathway and cisplatin resistance GSVA score difference by limma according to *SMARCA4* groups [32]. To interrogate oncogene scores, we additionally selected 41 oncogene signatures from MSigDB after excluding those with a low number of genes (<20), duplicates, or cancer type-specific pathways, such as those specific to lung and prostate cancer; the oncogene signatures’ GSVA scores were compared between two *SMARCA4* groups by the limma test [32]. Next, we also tested, by the oncogene signatures’ score difference, for SL and MSL types. In addition to patient meta-dataset analysis, we investigated downstream regulation of TNBC cell lines according to SMARCA4 activation and inactivation groups. We acquired the expression profiles of 19 TNBC cell lines, and classified these into activation and inactivation groups by median *SMARCA4* expression value (median = 6.69; Appendix A). EMT, cisplatin resistance, and oncogenic signal score were calculated similarly to previous meta-dataset analysis to refer equal signature gene sets. The difference in scores was tested by Wilcoxon rank sum test.

### 2.4. SMARCA4 Knockout Cells and Cisplatin Response

#### 2.4.1. Cell Culture and Reagents

The human TNBC cell lines MDA-MB-468 (HTB-132) and MDA-MB-231 (HTB-26) were obtained from ATCC (Manassas, VA, USA) and cultured in DMEM (Gibco, Grand Island, NY, USA) supplemented with 10% fetal bovine serum (Gibco, Grand Island, NY, USA) and 1% penicillin/streptomycin (15140-122; Gibco, Grand Island, NY, USA) at 37 °C and in 5% CO_2_. HEK293FT cells used for lentivirus production were grown and maintained in DMEM supplemented with 10% fetal bovine serum and 1% penicillin/streptomycin (Gibco, Grand Island, NY, USA). Cisplatin was purchased from Sigma-Aldrich (C2210000; St. Louis, MO, USA), and the YAP inhibitors (sitravatinib: S8573; crizotinib: S1068; verteporfin: S1786; CA3: S8661) were purchased from Selleckchem (Houston, TX, USA).

#### 2.4.2. Generation of Stable SMARCA4 KO Cell Lines Using CRISPR/Cas9

The following short guide RNA (sgRNA) targeting exon 4 of the human *SMARCA4* gene (NM_001128849) was designed on the CRISPR design website (https://design.synthego.com, accessed on 4 November 2019): forward: 5′-ATGGTCCCTCTCGCAGCCCA-3′, and reverse: 5′-TGGGCTGCGAGAGGGACCAT-3′. Then, 25 bp DNA oligos containing the 20 bp target sequence and the BsmBI sticky end were annealed and inserted into the lentiCRISPR-v2 plasmid (52961; Addgene, Watertown, MA, USA) digested with BsmBI (0580S; NEB, Ipswich, MA, USA). The resulting lentiCRISPR-v2-sgRNA plasmids were then cotransfected with the lentiviral packaging vectors pCMV-VSV-G and pCMV-dR8.2 into HEK293FT cells in Opti-MEM (31985-070; Gibco) using the Lipofectamine LTX Reagent (15338-100; Invitrogen, Carlsbad, CA, USA) to generate lentivirus particles. The viral supernatants were harvested 48 h after transfection and filtered through a 0.45 μm filter. Subsequently, MDA-MB-468 and MDA-MB-231 cells were transduced by the lentivirus for 48 h and then polyclonal *SMARCA4* KO pools were selected for at least 2 days with puromycin (A11138-03; Gibco, Grand Island, NY, USA) (2 µg/mL) containing DMEM medium. To confirm KO, genomic DNA from *SMARCA4* sgRNA-transduced cells was extracted using the DNA Mini Kit (51304; Qiagen, Valencia, CA, USA) according to the manufacturer’s instructions and amplified using primers (forward: 5′-TCCCGGGTGGTAGAAGATGA-3′, and reverse: 5′-CAGTAAGTTCCTGGGTGGCA-3′) for the sgRNA target site. The PCR cycling conditions were as follows: one cycle at 95 °C for 1 min, followed by 35 cycles of 95 °C for 15 s, 59 °C for 15 s, and 72 °C for 30 s with the final elongation step at 72 °C for 5 min. The PCR products were analyzed using tracking of indels by sequence decomposition (TIDE) [33].

#### 2.4.3. Cell Viability Analysis

Cell viability was assessed with the cell counting kit-8 (CCK-8; CK04; Dojindo Laboratories, Kumamoto, Japan). MDA-MB-468 and MDA-MB-231 cells were seeded in 96-well plates at 2.5 × 10^4^ and 1 × 10^4^ cells/well, respectively. After 20 h, the cells were treated for 48 h with DMSO (vehicle control), cisplatin (1–50 μM), sitravatinib (1–20 μM), crizotinib (1–20 μM), verteporfin (0.5–10 μM), or CA3 (0.1–4 μM). Then, 10% CCK-8 solution was added to each well, and cell viability was determined by measuring the absorbance at 450 nm with a microplate reader (VersaMax, Chester, PA, USA). In the reconstitution of *SMARCA4*, control and *SMARCA4* KO cells were transiently transfected with/without *SMARCA4* plasmid (65391; Addgene, Watertown, MA, USA) using Lipofectamine LTX reagent for 48 h, and cell viability of cisplatin was determined using CCK-8. Cell viability and IC50 µM values were calculated using dose response curves, plotted using GraphPad Prism 7.0 software (San Diego, CA, USA). 

#### 2.4.4. Quantitative Reverse Transcription PCR (RT-qPCR) and Immunoblot Analysis

Total RNA was isolated from MDA-MB-468 and MDA-MB-231 cells with RNeasy mini kit (74106; Qiagen, Hilden, Germany) according to the manufacturer’s instructions, and 1 μg RNA was converted to cDNA using the SuperScript™ IV Reverse Transcriptase cDNA Synthesis Kit (18090050; Invitrogen, Carlsbad, CA, USA). RT-qPCR assays were performed using SYBR Green I Master Mix (04707516001; Roche, Mannheim, Germany) and a real-time PCR system (05815916001; Roche). Glyceraldehyde 3-phosphate dehydrogenase (GAPDH) mRNA was used as a normalization control. RT-qPCR primers are shown in Appendix A. For immunoblot assays, cell lysates were prepared using TAP lysis buffer (25 mM Tris, pH 7.4, 140 mM NaCl, 0.5% NP-40, 10 mM NaF, 1 mM DTT, 1 mM PMSF, 1 mM EDTA, 1 mM Na_3_VO_4_, 1 mM β-glycerophosphate, 10% glycerol, and protease and phosphatase inhibitors), and protein concentration was determined by the Bradford assay (23225; Thermo, Waltham, MA, USA). Proteins were separated by SDS-PAGE and transferred to nitrocellulose membranes, which were incubated overnight at 4 °C with primary antibodies against SMARCA4 (ab110641; Abcam, Cambridge, UK) at 1:2000 dilution and β-actin (A5441; Sigma, St. Louis, MO, USA) at 1:5000 dilution or with the Epithelial-Mesenchymal Transition (EMT) Antibody Sampler Kit (9782; CST, Beverly, MA, USA)at 1:2000 dilution; horseradish peroxidase (HRP)-conjugated anti-rabbit IgG (111-035-144; Jackson ImmunoResearch, West Grove, PA, USA),and anti-mouse IgG (315-035-045; Jackson ImmunoResearch) at 1:2000 dilution were used as secondary antibodies, and the results were visualized and imaged using Fusion Solo 7S (Vilber Lourmat, Collegien, France).

#### 2.4.5. Flow Cytometry

MDA-MB-231 and MDA-MB-468 cells were detached by incubation with 0.2% trypsin-EDTA (15400-054; Invitrogen) for 1–3 min, washed with PBS containing 1% FBS, and incubated for 30 min at 4 °C with a phycoerythrin (PE)-conjugated EpCAM antibody (118215; Biolegend, San Diego, CA, USA). After a final wash, the cells were resuspended in 1% FBS in PBS and analyzed with a FACSCalibur machine (BD Biosciences, San Jose, CA, USA). FACS data were analyzed using FlowJo software version 10.6 (Tree Star Inc., Ashland, OR, USA).

#### 2.4.6. Cell Migration Assay and Wound-Healing Assay

Cell migration assays were performed by using 24-well plates containing cell culture inserts with 8 μm pore PET membrane Transwell chambers (3464; Corning Inc., Corning, NY, USA). Briefly, cells were plated at 1 × 10^5^ cells/well in 200 μL culture medium containing 1% FBS in the upper chamber of permeable Transwell supports, and cell migration was tracked for 24 h. Then, cells on the upper surface of the membranes were carefully removed with a cotton swab. Migration cells on the lower side of the membrane were stained with 0.1% crystal violet and then counted in three randomly selected fields per well under a light microscope. For the wound-healing assay, cells were seeded at a density of 5 × 10^4^ cells on both sides of an Ibidi culture insert (81176; Ibidi, Munich, Germany). Cells were permitted to attach and grow for 24 h, after which the insert was removed. The cell-free gap was photographed at 0, 24, 48 and 72 h with a ZEISS Axio Observer 7 microscope (Carl Zeiss, Jena, Germany) using a 10× objective.

#### 2.4.7. Immunofluorescence Assay

The human TMA (tissue microarray) paraffin sections of human breast carcinoma were purchased from US Biomax (BR802C; Biomax, Rockville, MD, USA). The tissue slide was deparaffinized in xylene (4322; BBC biochemical, McKinney, TX, USA) and rehydrated in a serial concentration of ethanol (from 100 to 80%), and rehydrated in PBS. Subsequently, tissue antigens were unmasked by heating in citrate buffer with pH 6.0 (C9999; Sigma-Aldrich, St. Louis, MO, USA). Tissue was permeabilized with 0.4 M glycine and 0.5% Triton X-100(X-100; Sigma-Aldrich), and nonspecific binding was blocked with 2.5% normal goat serum (S-1012-50; Vector Laboratories, Burlingame, CA, USA). The antibodies for tissues are as follows: SMARCA4 (dilution at 1:100), secondary antibody for immunofluorescence was Alexa Fluor 488 rabbit IgG (A-21206; Invitrogen, Carlsbad, CA, USA). Finally, sections were washed with PBS and mounted with VECTASHIELD mounting medium with DAPI (H-1200; Vector Laboratories). Then, samples were optically scanned using ZEISS Axio Observer 7 (Carl Zeiss, Jena, Germany)

### 2.5. RNA-Seq Analysis of SMARCA4 KO TNBC Cells

For RNA-seq analysis, *SMARCA4* KO-induced MDA-MB-468 and MDA-MB-231 cells were cultured at 70% confluency and then harvested with trypsin-EDTA. The cells were washed by centrifugation in PBS and re-suspended in 500 μL TRIzol reagent (15596018; Invitrogen, Carlsbad, CA, USA) at a concentration of about 1 × 10^6^ cells/mL. Then, cells were sent to Macrogen (Macrogen Inc, Seoul, South Korea) for RNA isolation and library preparation/sequencing.

To extract gene expression profile, we trimmed the reads using Trimmomatic, and STAR (version 2.5.2a) in two-pass mapping mode was used to align reads against the hg19 reference genome [34]. Next, we generated gene expression profiles using RSEM v1.1.13 [35]; genes with low expression or low variability (average FPKM < 1 or SD < 1) were eliminated. Differentially expressed genes (DEGs) between wild-type (WT) and *SMARCA4* KO cell lines were extracted by limma with adjusted *p*-value (<0.05) and log-scaled fold-change (FC > 0.5) cutoffs. Pathway gene set enrichment analysis was performed using the Cytoscape ReactomeFIViz plug-in [36]. Hippo pathway and YAP/TAZ target gene signatures were collected from the literature [37]. Next, we computed YAP/TAZ scores by the average gene expression of signature gene set. The YAP/TAZ score differences between the KO and WT cell lines were tested by Welch’s *t*-test.

## 3. Results

### 3.1. Progression-Associated Driver Gene Variants Identified in Samples from TNBC Patients Treated with Neoadjuvant Cisplatin

To understand the driver variants and their biological process in refractory TNBC, progression-associated driver genes were identified from our TNBC cohort and large-scale TNBC cohorts [19,20,22,38]. We extracted mutation genes that increased in cisplatin nonresponder and metastasis patients, and clarified concurrent pathways of mutation genes. First, we analyzed WES from 37 advanced refractory TNBC patients (Appendix A) who received cisplatin therapy, and identified somatic mutations and copy number variants. We extracted variants of 116 genes previously identified in breast cancer and pan-cancer driver gene studies [39]. *TP53* (76%) and *PIK3CA* (24%) were the most recurrently mutated genes identified in our TNBC cohort, and *ATM*, *BRCA1* and *NF1* mutations were present in more than 10% of the samples (Figure 1A). *MYC* and *RAD21* amplifications were identified (22%; peak at 8q23.1, q-value = 2.14 × 10^−5^), as were *CDKN2A* deletions (8%; peak at 9p21.3, q-value = 0.003). Next, we investigated mutations at the pathway level, after classifying the patients by cisplatin response (nonresponder *n* = 14 and responder *n* = 20). The mutations of genes in the PI3K signaling pathway were significantly enriched in the responder group (chi-square test *p* = 0.078; Figure 1A), whereas the mutations in genes related to the SWI/SNF complex (*p* = 0.056) and protein homeostasis/ubiquitination (*p* = 0.007) were more frequent among nonresponders (Figure 1A). Interestingly, *SMARCA4* was the most frequently mutated gene (*n* = 3) that was unique to nonresponders in our dataset.

To validate the gene mutation findings in our small cohort of TNBC patients, and to ascertain whether there is an overlap among the genes implicated in metastasis and cisplatin response, we analyzed gene mutation frequencies in four large-scale breast cancer cohorts including primary and metastatic samples (*n* = 3455) [19,20,22,38]. TP53 mutations were present in a high proportion of metastases (63.4%), but were also quite dominant in primary tumors (50.5%; odds ratio (OR) for metastases/primary = 1.32). *NF1*, *STAG2*, *ATM* and *SMARCA4* mutations (chi-square test *p* < 0.01, OR > 1.68) were remarkably increased in metastases compared to primary samples (Figure 1B). *PIK3CA* mutations (OR = 0.97) were less common in metastatic TNBC than in primary TNBC (Figure 1B). Among the five genes related to epigenetic regulation in our analysis (Figure 1B, purple dots), *ARID2* (OR = 2.32) and *SMARCA4* (OR = 1.90) mutations showed an association with metastasis. Interestingly, these chromatin SWI/SNF-related genes demonstrated progression-free survival (PFS), and patients (*n* = 8) harboring SWI/SNF gene mutations exhibited a shorter progression time than wild-type patients (HR = 2.45 and PFS *p* = 0.09; Figure 1C). Also, there were significant differences in the PFS of our TNBC patients according to cisplatin response (HR = 3.97 and PFS *p* = 0.01; Figure 1C). This suggests a relationship between the SWI/SNF pathway mutation and cisplatin resistance that influences patient outcomes.

### 3.2. Transcriptional Characteristics and Subtypes of TNBC Are Regulated by SMARCA4

Given the association of SWI/SNF pathway mutations with cisplatin response, we aimed to further characterize the TNBC samples in terms of the activity of this pathway. To accomplish this, we analyzed the gene expression profiles (*n* = 957) from Gene Expression Omnibus (GEO), classified by molecular subtype (MSL, LAR, IM, and SL), based on machine learning methods [7]. For this analysis, we focused on *SMARCA4* as the most recurrent mutation gene in our TNBC cohort, and as relatively unknown compared to *ARID1A* and *ARID2*. The somatic mutation of *SMARCA4* leads to LoF, and its down-expression concurrently plays the same role in the tumor [11,12,13]. Therefore, we performed gene expression analysis according to *SMARCA4* expression. First, we classified the TNBC samples into activation and inactivation groups based on *SMARCA4* gene expression, and investigated the association with TNBC subtype. The *SMARCA4* activation group was enriched in the SL subtype, whereas the inactivation group was enriched in the MSL subtype (Fisher’s exact test *p* = 1.83 × 10^−24^, *SMARCA4* log2 FC = 1.32 between MSL and SL; Figure 2A). The AR and IM subtypes were interspersed among the samples with intermediate *SMARCA4* expression.

*MYC*, cell cycle-related pathways, and cancer stemness are associated with the SL subtype, and EMT and TGF-β pathway activation is implicated in the MSL subtype [7,8,9,10]. Consequently, cisplatin nonresponders showed over-representative of both EMT activation and the MSL subtype [7], so we also considered the cisplatin resistance score next. We verified the signature scores of the EMT for the MSL subtype and stemness for the SL subtypes, respectively, and found significant differences (EMT *p* = 8.67 × 10^−33^, stemness *p* = 7.30 × 10^−14^) depending on the *SMARCA4* activation status (Figure 2B). Moreover, *SMARCA4* inactivation was associated with cisplatin resistance (*p* = 2.2 × 10^−8^; Figure 2B). To further verify these associations, we looked deeper at the biological processes enriched in the two *SMARCA4* groups. The global biological processes identified in each *SMARCA4* group corresponded to pathways that were representative of the MSL and SL subtypes (Figure 2C); the apoptosis, myogenesis and EMT pathways activated in the MSL subtype were enriched in the *SMARCA4* inactivation group, whereas typical SL processes, such as *MYC* targets, G2M checkpoint, and DNA repair, were upregulated in the *SMARCA4* activation group (Figure 2C) [7]. In the investigation categorized by *SMARCA4* activity, we consistently verified that *SMARCA4* inactivation is associated with the MSL subtype, in EMT activation and cisplatin resistance.

To identify targetable oncogenes, we estimated the activity of 41 oncogene signatures (Materials and Methods), and tested their ability to discriminate *SMARCA4* activity and TNBC subtype. The activation of *CTNNB1*, *LEF1*, *TGF-β*, and *YAP1* signatures was over-represented in the MSL (*p* < 2.85 × 10^−4^) and *SMARCA4* (*p* < 1.58 × 10^−6^) inactivation group (Figure 2D); *CTNNB1*, *TGF-β*, and *YAP1* promote cell invasion [40] to implicate the MSL subtype and *SMARCA4* inactivation. In contrast, the *E2F1*, *MYC* and mTOR pathways were enriched in both the SL subtype and *SMARCA4* activation groups (Figure 2C,D). The cisplatin resistance score [7] showed a positive correlation with EMT activation (*p* = 9.89 × 10^−26^, *R* = 0.45; Appendix A). Lastly, previously identified oncogenic signatures, including *LEF1*, *TGF-β*, *TBK1*, and *YAP1*, showed positive correlations with cisplatin resistance (Appendix A). These results of patient meta-expression profile analysis were consistent in TNBC cell lines (*n* = 19). The cell lines were classified into SMARCA4 inactivation (*n* = 11) and activation (*n* = 8) by the median mRNA expression (Appendix A). Unsurprisingly, the EMT pathway was activated in the inactivation group (*p* = 0.12; Appendix A) and cisplatin resistance increased in the inactivation group (*p* = 0.13). The oncogenic signals *VEGF*, *YAP1*, *TGF-β1*, and *LEF1* were also activated in the inactivation group (*p* < 2.49 × 10^−22^). In summary, *SMARCA4* inactivation is involved in *LEF1*, *TGFR1*, and *YAP1* oncogenic signal activation, the MSL subtype among four subtypes, and the resistance of cisplatin in TNBC.

### 3.3. SMARCA4 Knock-Out to Evaluate Cisplatin Resistance

Given the potential implication of *SMARCA4* inactivation to induce the MSL subtype in TNBC and the therapeutic response to cisplatin, we aimed to evaluate the downstream molecular response to *SMARCA4* inactivation by using the CRISPR/Cas9 system to knockout (KO) *SMARCA4* in TNBC cell lines. We selected two different human breast cancer cell lines, MDA-MB-468 and MDA-MB-231, for these functional experiments; MDA-MB-468 cells are classified as SL, and MDA-MB-231 cells represent the MSL subtype [10]. To verify successful interruption of the *SMARCA4* genomic sequence, we sequenced *SMARCA4* gRNA-specific amplicons in genomic DNA derived from MDA-MB-468, and its *SMARCA4* KO cells. As expected, increased aberrations in *SMARCA4* KO gRNA target sequences were observed in *SMARCA4* KO cells relative to the control cells (Figure 3A). Next, we verified that CRISPR/Cas9-mediated *SMARCA4* KO considerably decreased *SMARCA4* protein expression in both MDA-MB-468 and MDA-MB-231 cells (Figure 3B). To assess the effect of *SMARCA4* KO on cisplatin sensitivity, the cells were exposed to various concentrations of cisplatin (1–50 μM) for 48 h, and then a viability evaluation was performed. MDA-MB-468 *SMARCA4* KO cells exhibited the highest cisplatin resistance, with a 1.5-fold higher IC_50_ than the control cells (Figure 3C). The IC_50_ for cisplatin was higher in the MSL-type MDA-MB-231 cells than in the MDA-MB-468 cells (Figure 3C). To further validate the function of *SMARCA4*, we rescued *SMARCA4* by overexpressing it in *SMARCA4* KO cells. As expected, we observed a reconstituted exogenous and endogenous protein level of SMARCA4 in both MDA-MB-468 and MDA-MB 231 cells (Figure 3C, right). The results showed that overexpressed *SMARAC4* in *SMARCA4* KO attenuates the resistance to cisplatin in both MDA-MB-468 and MDA-MB-231 *SMARCA4* KO cells (Figure 3C, left). These data suggest that *SMARCA4* LoF promotes cisplatin resistance to varying degrees, depending on the molecular characteristics of the TNBC cells.

### 3.4. SMARCA4 KO Transforms Basal-like MDA-MB-468 Cells to an EMT Activated Cell Line

To pinpoint the pathways affected by *SMARCA4* LoF, we explored global gene regulation in *SMARCA4* KO cells by DEG analysis (adj. *p* < 0.05 and log2 FC > 0.5; limma test) from three replicate KO samples compared to controls for each cell line. We identified 156 DEGs between the MDA-MB-468 cell groups and 289 DEGs between the MDA-MB-231 cell groups (Figure 4A). Two upregulated DEG sets in the KO cell lines denoted the activation of the following EMT-associated pathways: ECM organization and Hippo signaling pathways (FDR < 0.05; Appendix A). In particular, the *TGF*-*β**1*, *PIM1*, *CDKN1A* and *BIRC3* genes, which promote EMT, were upregulated by more than two-fold in the MDA-MB-468 KO cells compared to the control cells (Figure 4A and Appendix A). Additionally, the expression of an EMT signature also presented the increase in only MDA-MB-468 KO (FC = 1.23; Figure 4B) [31]. As MDA-MB-231 cells, which are characterized as the MSL subtype, showed no drastic change (FC = 0.93) in EMT gene expression after KO, we concluded that the global expression of genes in this pathway is important for maintaining MSL characteristics.

We inferred that MDA-MB-468 KO undergo a transition from the SL to MSL subtype due to changes in the EMT pathway. Therefore, we additionally interrogated the EMT pathway by assessing changes in the mRNA and protein expression. As expected, the mRNA expression of EMT signature genes, including Snail, Twist, N-cadherin, Slug, and Claudin-1, was increased in MDA-MB-468 *SMARCA4* KO more than the control (*p* < 0.05; Figure 4C). Consistently, immunoblotting showed increased Snail, Claudin-1, N-cadherin and E-cadherin expression in *SMARCA4* KO MDA-MB-468 cells compared with control cells (Figure 4D). In contrast, no significant changes were observed in the expression of three proteins in MDA-MB-231 KO, and N-cadherin and Slug even decreased (Figure 4C,D).

To confirm the subtype switch in MDA-MB-468 KO from WT, we performed FACS analysis to assess the epithelial cell population as determined by EpCAM positivity. The MDA-MB-468 cell population clearly had a higher proportion (15.8%) of EpCAM-positive (epithelial) cells than the MSL-type MDA-MB-231 (1.46%) (Figure 4E). In the *SMARCA4* KO groups, EpCAM positivity markedly decreased among MDA-MB-468 cells (control: 15.8%, KO: 2.01%), but showed no considerable change among MDA-MB-231 cells (control: 1.46%, KO: 3.29%; Figure 4E). To confirm this molecular change using functional assays, we evaluated the migration and invasion according to *SMARCA4* KO. MDA-MB-468 cell migration, as determined by wound-healing assays, increased in *SMARCA4* KO in a time-dependent manner (0 to 72 h) (Figure 4F). Consistently, the migration ability across a Transwell membrane was increased in *SMARCA4* KO MDA-MB-468 cells more than the control (Figure 4G). Additionally, to confirm the clinical and biological significance of *SMARCA4* in breast cancer patients, we compared the expression profiles of *SMARCA4* in non-invasive breast cancer tissue versus invasive breast cancer tissue. Immunofluorescence staining results revealed that *SMARCA4* expression was decreased to a greater extent in invasive breast cancer tissues compared with that in non-invasive breast cancer tissues (Appendix A). These results indicate that *SMARCA4* LoF induces EMT, invasiveness, and a change in subtype from SL to MSL.

### 3.5. Hippo Pathway Activation in Response to SMARCA4 KO and the Response to YAP/TAZ Inhibitor

The Hippo pathway targets *YAP*, promotes EMT, and enhances in vitro invasion [41]. In a previous KO cell line expression analysis, we identified extracellular matrix organization and Hippo signaling pathways (Appendix A). *YAP* was also identified as a regulator of the MSL type in the TNBC patient meta-expression analysis (Figure 2D and Appendix A). Thus, disrupting *YAP* function has the potential to decrease metastatic burden, so it presents a promising target [41]. Therefore, we further evaluated the Hippo-YAP/TAZ pathway activity and the sensitivity of YAP inhibitors. Basal expression of Hippo pathway components and YAP/TAZ target genes was maintained in MDA-MB-231 cells upon *SMARCA4* KO (*p* = 0.5; Figure 5A), whereas significant increases were observed in MDA-MB-468 cells (Hippo pathway, *p* = 0.2; YAP/TAZ, *p* = 0.09; Figure 5A). This implies that the transcriptional change in SL-type cells’ EMT genes is more dramatic than the MSL type. We could infer that there is an alteration of TNBC subtype from SL to MSL. From YAY/TAZ target genes identified from a previous study [37], nine target genes were chosen for an evaluation of up-regulation in *SMARCA4* KO MDA-MB-468 cells. *IGFBP3*, *NT5E*, *GADD45A*, and *TGF-β1* mRNA expression was drastically upregulated in *SMARCA4* KO cells compared with control cells (FC > 1.5) (Figure 5B). These results suggest that *SMARCA4* LoF induces the activation of Hippo-YAP/TAZ signaling.

To explore the targetability of the Hippo-YAP/TAZ signal, we examined the cell inhibitory effects of four YAP pathway inhibitors. Sitravatinib and crizotinib inhibit the phosphorylation of multiple receptor tyrosine kinases (RTKs), including YAP-AXL [42,43], and verteporfin and CA3 were selective inhibitors of the YAP/TEAD cascade [44]. The IC_50_ values of sitravatinib and crizotinib were not different between *SMARCA4* KO and control MDA-MB-468 cells (Figure 5C). Interestingly, verteporfin (IC_50_ FC = 1.35) and CA3 IC_50_ (FC = 1.24) were more sensitive to cancer cell viability in the KO group than the control group (Figure 5C). We performed the next studies using verteporfin, an FDA-approved drug. We aimed to determine whether verteporfin inhibits the *SMARCA4* KO-mediated YAP/TAZ target genes and the EMT pathway. The expression levels of the YAP/TAZ target genes and EMT were consistently increased in MDA-MB-468 *SMARCA4* KO cells compared with the control, and verteporfin suppressed the YAP/TAZ target genes and EMT pathway (Figure 5D). We also conducted normal human fibroblast cell-based tests to determine the human-specific toxicity of verteporfin. As expected, verteporfin-associated toxicity was not observed in the range of our experiment, which suggests that it does not have a response in breast cancer cells (Appendix A). As a functional readout, Transwell migration assays showed that verteporfin abolished *SMARCA4* KO MDA-MB-468 cell migration (Figure 5E). Taken together, our findings suggest that the *SMARCA4*-YAP cascade may facilitate cisplatin resistance, and regulate YAP target and EMT-related genes.

## 4. Discussion

Cisplatin resistance is a critical problem to TNBC treatment, so 22 clinical trials have been proposed [45]. Here, we revealed the clinical importance and biological role of *SMARCA4* in the cisplatin resistance of TNBC patients. *SMARCA4*, similarly to other oncogenes, plays dual roles of a dichotomy according to the activation status [17]. In our result, the LoF status of *SMARCA4* mutations was also over-representative in metastasis. In the transcriptome, *SMARCA4*-inactivated TNBC patients exhibited the MSL subtype. Consequently, the EMT pathway and oncogenic signals of *CTNNB1*, *LEF1*, *TGFβ*, and *YAP1* were activated in *SMARCA4* inactivation patients. Therefore, we concluded that *SMARCA4* depletion induces EMT upregulation and metastasis in TNBC. We also evaluated the findings from cell line experiments, then observed the cell invasiveness, migration, and the Hippo-YAP/TAZ signal upregulation. Especially, we demonstrated the ability of the YAP inhibitor verteporfin to suppress YAP/TAZ target genes. It was suggested as a therapy for cisplatin-resistant TNBC patients. In another study of *SMARCA4,* which involved an additional regulator and down-stream signals, SOX4 was revealed to cooperate with SMARCA4, to regulate PI3K-Akt signaling, and to target *TGFBR2* [46]. This finding was relevant with our result (Figure 2D) to extract *TGF-β* signal activation, and, additionally, suggested that PI3K is regulated by *SMARCA4*. Our histologic and molecular findings provide a novel insight into prognostic indicators according to *SMARCA4,* and therapeutic predictors for TNBC patients.

*TP53*, *NF1*, *ARID1A* and *KMT2C* mutations increased in metastasis compared with primary TNBC [22]. However, 58% of patients with metastatic breast cancer in this dataset had no actionable target [21]. Among the highly mutated driver genes are the epigenetic regulator *ARID1A*, which is an SWI/SNF family member whose inactivation leads to endocrine therapy resistance and stem-like (SL) features in estrogen receptor-positive breast cancer [47], and a LoF variant of the histone methyltransferase *KMT2C*, which reduces PFS and sensitivity to antiestrogen therapy [48]. *ARID1A*, *ATR* and *KDM6A* mutations’ epigenetic roles are relatively well known [47,48]. However, *SMARCA4* mutations were relatively unknown in TNBC. We reveal that *SMARCA4* exhibited invasiveness and cell migration by upregulation of the Hippo-YAP/TAZ signal in TNBC.

Hippo-YAP/TAZ signaling is a key factor in tumor progression and metastasis; in particular, *YAP1* promotes the EMT and invasion of breast epithelial cells [49,50]. However, *YAP1* amplification and gain-of-function mutations are rarely detected in pan-cancer studies [37,41]. Therefore, it is inadequate for an upstream marker to identify the Hippo-YAP/TAZ signal. In this study, we found that *SMARCA4* is an upstream suppressor of Hippo-YAP/TAZ signaling associated with cisplatin resistance in TNBC. Additionally, among the YAP/TAZ targets, *IGFBP3*, *NT5E*, *GADD45A*, *TGF-β1*, and *TGF-β2* showed the largest increases in expression FC (Figure 5B). Interestingly, the following target genes were the known prognostic factors, and support EMT, progression, and metastasis: *IGFBP3* (recurrence-free survival *p* = 0.0068), *NT5E* (OS *p* = 0.011), and *GADD45A* (OS *p* = 0.041) [51,52,53]. Although the prognosis of previous studies was investigated, functional studies using large-scale cohorts were not fully performed by *SMARCA4* or YAP/TAZ target genes, as well as cisplatin resistance.

To propose targeted therapy for MSL-type TNBC harboring SMARCA4 downregulation, we evaluated four YAP1 inhibitors. Sitravatinib and crizotinib, two multi-RTK inhibitors that inhibit the YAP–AXL axis, among others, are used extensively to treat cancer [54,55]; unfortunately, they showed no effective response in *SMARCA4* KO cells, whereas verteporfin and CA3, which decrease YAP1 expression [56,57], exhibited activity in this context. Verteporfin showed the greatest sensitivity, suppressing the expression of the *YAP1* target genes *IGFBP3*, *NT4E*, and *GADD45A*, and decreasing cell invasiveness. Verteporfin has been evaluated in clinical trials of various cancer types, including cutaneous metastatic breast cancer, and in vitro and in vivo experiments in human glioblastoma stem cells and xenograft models identified verteporfin as a therapeutic candidate in EGFR-variant glioblastoma [58]. Therefore, we propose verteporfin as a promising therapeutic agent for metastatic or refractory TNBC with *SMARCA4* LoF.

Our evaluation of *SMARCA4*, identified from a large-scale clinical dataset analysis, was performed by an in vitro test, and lacked in vivo testing. Nevertheless, previous studies revealed that the loss of *SMARCA4* resulted in increases in both the number and size of tumors when compared with controls [59]. Also, *SMARCA4*-null heterozygous mice spontaneously develop mammary tumors, and specific alleles of *SMARCA4* are associated with cancer predisposition in humans [60]. Furthermore, verteporfin reduced the expression of *YAP* in the ZNF367-overexpressing breast cancer MDA-MB-231 and 4T1 cells, and significantly reduced lung metastases in mouse models [61]. This finding provided indirect support to our finding in the clinical data and cell culture model. However, our result primarily focused on the clinical and biological role of *SMARCA4* from multiple resources, such as mutation and gene expression profiles from large-scale cohorts. Therefore, we are convinced that our data provide good evidence for the role of *SMARCA4* inactivation in exhibiting metastasis and cisplatin resistance for the induction of transcriptome changes from the SL to the MSL subtype in TNBC, and can, thus, be a good origin for further animal studies.

Our studies investigated the clinical and biological role of *SMARCA4* from multiple resources, such as mutation and gene expression profiles from large-scale cohorts. To hypothetically sum up, the depletion of *SMARCA4* promotes the transcription factor YAP/TAZ to regulate the activation of EMT genes in TNBC cells (Figure 6). Verteporfin selectively inhibited *YAP*, and EMT genes were consequently down-regulated. Unfortunately, our experiment includes limitations in the usage of restricted TNBC cell lines without epigenomic evidence from ATAC-seq or ChIP-seq. In spite of these limitations, we employed the representative TNBC cell lines for the MSL and SL subtypes. The tumor-associated functions are better explained than in the previous experiment using a mammalian epithelial cell line [18]. However, it will improve our observation to use the extensive cell lines in our next experiment. Currently, the transcriptional change in our KO cell line was investigated from only gene expression profile. Additional sequencing approaches could explain more details, such as chromatic organization or transcription factor binding by *SMARCA4* deficiency. In clinical aspects, the pathologic and therapeutic profile should be supported to prove the oncogenic role by *SMARCA4* deficiency.

## 5. Conclusions

In summary, our studies reveal the clinical and therapeutic importance of *SMARCA4* depletion to exhibit metastasis and cisplatin resistance in TNBC. We successfully identified EMT and Hippo-YAP/TAZ signal regulation by *SMARCA4* depletion. Following our finding, *SMARCA4* could be an important regulator and a diagnostic marker in TNBC. Further, the expansion of clinical evaluations and survival analyses is required. The advanced therapeutic strategy using the YAP inhibitor also remains for a better outcome. We believe that our finding improves the diagnosis and therapy for TNBC patients.

## Figures and Tables

**Figure 1 cancers-13-05474-f001:**
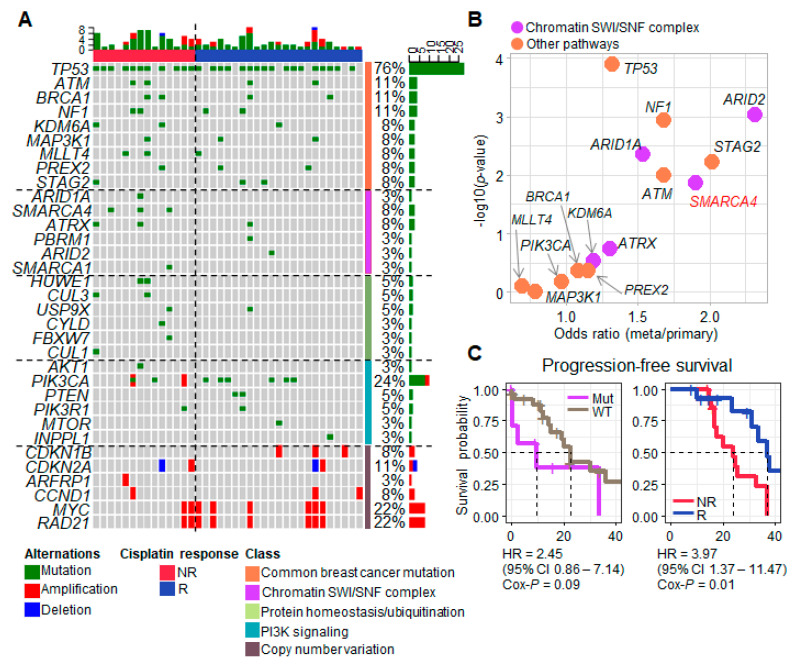
Variant landscape of TNBC according to cisplatin response. (**A**) An oncoplot of variant genes classified by the response of triple-negative breast cancer (TNBC) patients (*n* = 37) to cisplatin (NR, nonresponder; R, responder). Driver genes were categorized by oncogene pathways (37). (**B**) A scatter plot of mutated genes in metastatic TNBC. The X axis is the odds ratio (OR) of metastatic-to-primary TNBC, and the Y axis is the *p*-value calculated by Fisher’s exact test from the contingency table of metastasis and gene mutation cases. (**C**) Progression-free survival (PFS) curves of the TNBC patients who had mutation of chromatin SWI/SNF complex genes and cisplatin response clinical profile (NR: nonresponder; R: responder), respectively. The hazard ratio (HR) and *p*-value were calculated by Cox proportional regression analysis.

**Figure 2 cancers-13-05474-f002:**
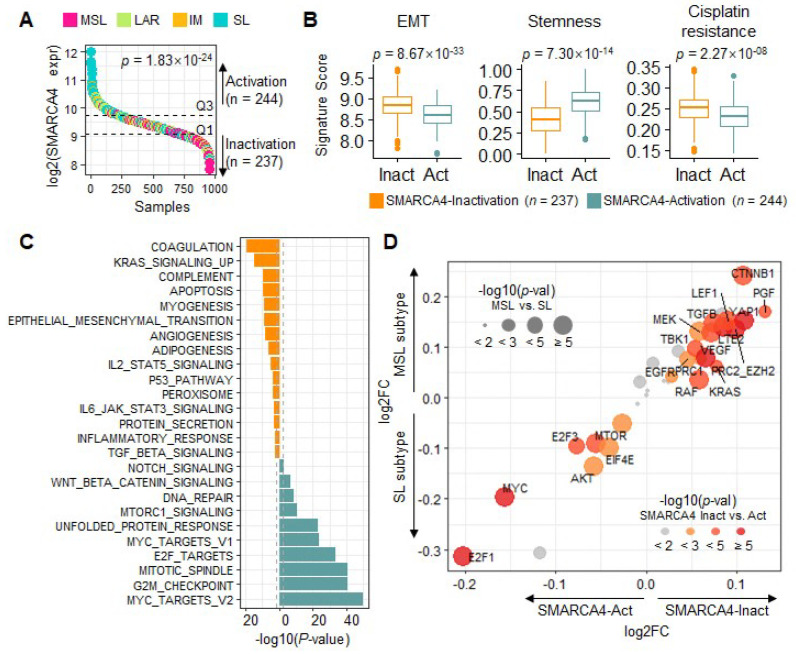
Global gene regulation according to *SMARCA4* activation status. (**A**) A waterfall plot of sorted *SMARCA4* gene expressions of TNBC samples (*n* = 957). Y-axis indicates log2-scale *SMARCA4* mRNA expression. Each sample is colored according to TNBC subtype (MSL: pink; LAR: light green; IM: orange; SL: light blue). The *p*-value denotes the enrichment of subtypes by *SMARCA4* activation group as determined by chi-square analysis. (**B**) Boxplots of gene signature activity scores in the *SMARCA4* inactivation (orange) and activation (teal) groups. Two groups were classified by upper and lower tertile of *SMARCA4* gene expression. The EMT and stemness scores are indicative of the MSL and SL subtypes, respectively, and the cisplatin resistance score is also shown. *p*-values were calculated by *t*-tests. (**C**) Bar plot of log-scale *p*-values for GSVA limma test for the *SMARCA4* inactivation (orange) and activation (teal) groups. (**D**) Scatter plot of the oncogene signature scores. The X- and Y-axes indicate the log2FC in the average scores between the *SMARCA4* activation and inactivation groups and between the MSL and SL groups, respectively. Circle size indicates the rescaled *p*-value for the GSVA of oncogene signatures between subtypes (MSL vs. SL), and circle color indicates the rescaled *p*-value according to *SMARCA4* group (inactivation vs. activation).

**Figure 3 cancers-13-05474-f003:**
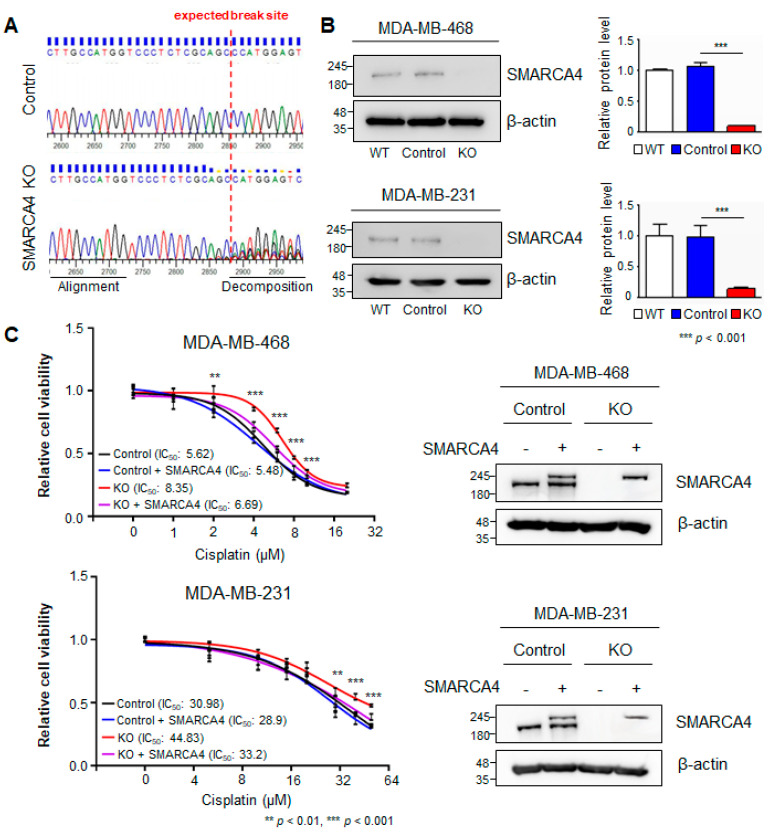
Effect of *SMARCA4* KO on the cisplatin response of malignant breast cancer cells. (**A**) Sequence-based validation of genome editing using *SMARCA4* sgRNA with the CRISPR/Cas9 system in MDA-MB-468 cells. (**B**) Western blot analysis of SMARCA4 protein expression in parental (WT) and genome-edited cell lines exposed to the CRISPR/Cas9 nontargeting control vector (control) or CRISPR/Cas9 *SMARCA4* KO vector (KO). Representative Western blots from three independent biological replicates are shown. Bar plots show SMARCA4 protein expression for WT, control and KO. (**C**) Cisplatin drug response curves after 48 h in both MDA-MB-468 and MDA-MB-231 cells. Cell viability was examined under the following conditions: control or *SMARCA4* KO or in combination with SMARCA4. Exogenous SMARCA4 expression in these rescued cell lines was analyzed by Western blot. β-actin was used as the loading control. All experiments were performed in triplicates, and the data have been presented as mean ± SD. Statistical analyses were carried out using Graph Pad Prism v7.0, data were analyzed by unpaired *t*-test; ** *p* < 0.01, *** *p* < 0.001.

**Figure 4 cancers-13-05474-f004:**
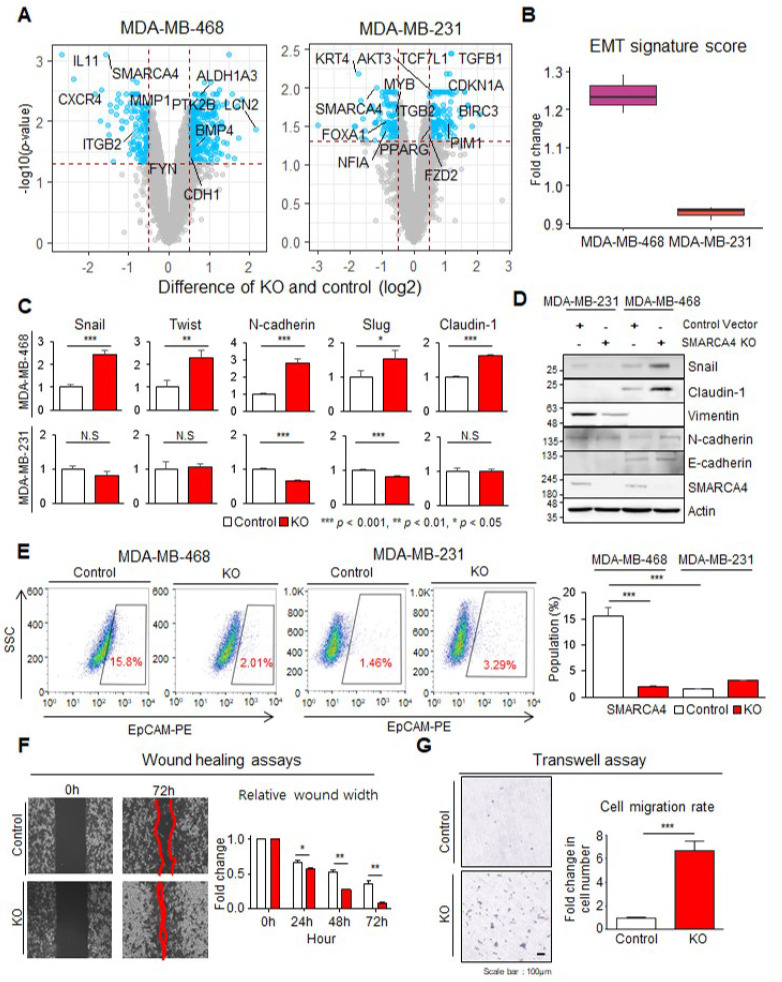
EMT characteristics induced by *SMARCA4* KO. (**A**) Volcano plots of DEGs based on RNA-seq analysis of control and *SMARCA4* KO cells. The X-axis indicates the log2FC between *SMARCA4* KO and control cells. The Y-axis is the −log10-scaled *p*-value calculated using the limma test. DEGs involved in EMT and the *TGF-β* pathway are labeled. (**B**) Boxplot of the fold change (FC) in EMT signature score between the KO and control versions of MDA-MB-231 and MBA-MD-468 cells. EMT scores were calculated from signature genes (*n* = 42) [31], and fold changes in scores were acquired to compare WT and KO for two cell lines. (**C**) The mRNA expression of the EMT markers Snail, Twist, N-cadherin, Slug, and Claudin-1 was calculated using real-time PCR data for both control and *SMARCA4* KO MDA-MB-468 and MDA-MB-231 cells; the expression value in the control group was set to 1. The *p*-values were indicated by each asterisk. (**D**) Immunoblot analysis of EMT (Snail, Claudin-1, Vimentin, and N-cadherin) and epithelial (E-cadherin) marker expression levels in *SMARCA4* KO and control cells. (**E**) Flow cytometry analysis of EpCAM expression in MDA-MB-468 and MDA-MB-231 cells. The percentage of PE-positive cells was determined from analysis of 10,000 events. Data are the average of 3 independent experiments performed and presented in a bar plot. (**F**) Relative wound width in control and *SMARCA4* KO MDA-MB-468 cells. The bar plot shows the FC in migration distance after gap distance at 0 h was set to 1. Representative images of three independent experiments are shown on the left. (**G**) Transwell migration of MDA-MB-468 control and KO cells. Representative images are shown on the left (scale bar, 100 µm). Cell migration events across the membrane were stained and counted. Moreover, FC in cell migration compared with control is presented in the bar plot (right). All experiments were performed in triplicates, and the data have been presented as mean ± SD. Statistical analyses were carried out using Graph Pad Prism v7.0, and data were analyzed by unpaired *t*-test; * *p* < 0.05, ** *p* < 0.01, *** *p* < 0.001, N.S. = not significant.

**Figure 5 cancers-13-05474-f005:**
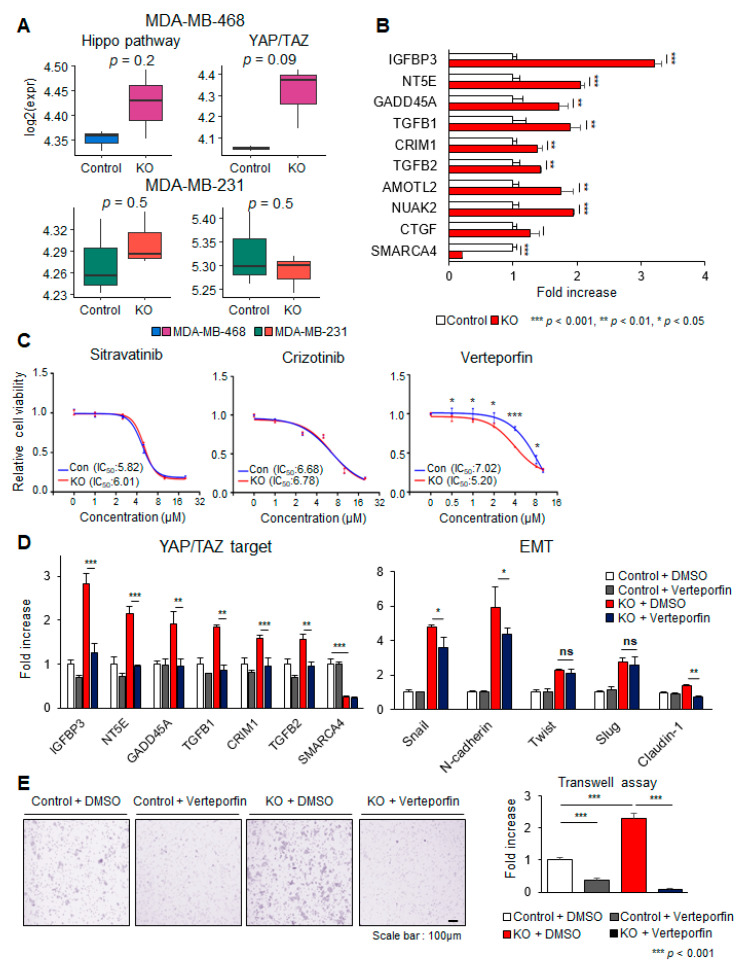
DEG analysis to identify potential therapeutic targets in *SMARCA4* KO cells. (**A**) Boxplots of the log2 gene set scores of the Hippo pathway gene set (*n* = 19) and YAP/TAZ target gene set (*n* = 22) in control and *SMARCA4* knockout (KO) MDA-MB-468 and MDA-MB-231 cells. *p*-values were calculated using Welch’s *t*-test. (**B**) Bar plot of *YAP* target gene expression. The fold change (FC; Y-axis) in mRNA expression of each gene (Y-axis) was determined by real-time PCR analysis of MDA-MB-468 control and KO cells. (**C**) Drug response curves based on the viability of MDA-MB-468 *SMARCA4* KO and control (Con) cells treated with the YAP/TAZ inhibitors sitravatinib, crizotinib, verteporfin and CA3. The IC_50_ for each drug in each cell line is provided in each panel. (**D**) FC in the gene expression of YAP/TAZ target (left) and EMT marker (right) genes in *SMARCA4* KO and control MDA-MB-468 cells before and after treatment with verteporfin (1 μM, 48 h) or DMSO. (**E**) Transwell migration assays according to verteporfin and DMSO treatment for *SMARCA4* KO and control. The fold increase in the number of migrated cells is presented in the bar plot. The *p*-values were indicated by each asterisk. All experiments were performed in triplicates, and the data have been presented as mean ± SD. Statistical analyses were carried out using Graph Pad Prism v7.0, and data were analyzed by unpaired *t*-test; * *p* < 0.05, ** *p* < 0.01, *** *p* < 0.001, N.S. = not significant.

**Figure 6 cancers-13-05474-f006:**
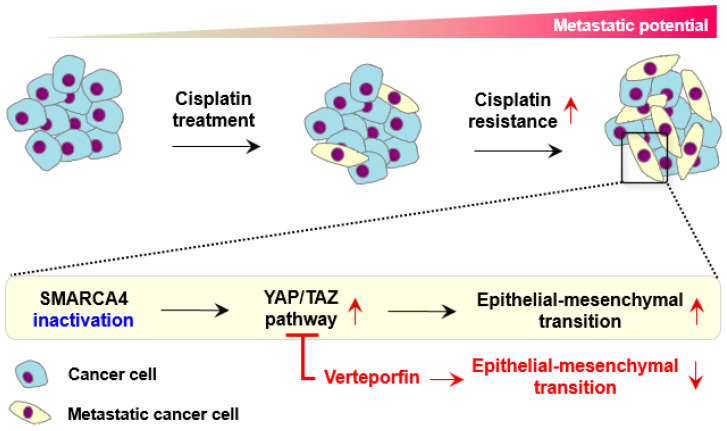
Hypothesis illustration to describe EMT regulation by SMARCA4 deficiency, and verteporfin to inhibit YAP in TNBC.

## Data Availability

The next-generation sequencing files have been deposited in NCBI SRA accession SRP156081, and *SMARCA4* KO RNA-seq files have been deposited in NCBI GEO accession GSE178532.

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
