# Peer review of "SMARCA4 Depletion Induces Cisplatin Resistance by Activating YAP1-Mediated Epithelial-to-Mesenchymal Transition in Triple-Negative Breast Cancer"

_cancers, 2021, doi:10.3390/cancers13215474_

Round 1

Reviewer 1 Report

In the present research, authors employ large-scale genome and transcriptome profiles to examine clinical implications of SMARCA4 inactivation. The results were then validated in two cell lines to identify the pathway details. SMARCA4 mutations were observed in patients with cisplatin resistance and associated with high metastasis. As the SMARCA4-knockout cells showed activation of the Hippo pathway in turn driving cisplatin resistance, the authors propose to employ YIP inhibitors to rescue the resistance.

I have several reservations; my comments are appended as below:

Major comments:

  1. Introduction section- please discuss the metastasis and prognosis in TNBC patients and patients with cisplatin resistance.
  2. What cutoff do authors use to divide samples by SMARCA4 mRNA expression into inactivation and activation groups?
  3. What were the P-value and fold-change (FC) parameters to analyze the RNAseq data?
  4. How responders and nonresponders to cisplatin treatment were defined?
  5. Figure 2B- please include n
  6. Figure 2-Unless validated in an experimental setting, it's hard to judge the credibility. Authors can try to validate the candidate targets in cell lines/human samples.
  7. Figure 3- do authors use single sgRNA or in the mixture?
  8. Unless I am missing, the EMT gene signature score in figure 4B is based on figure 4A. How many genes does it include (used checked in figure 4B)?
  9. As authors use only cancer cell lines, how do they rule out the toxic effects of inhibitors on normal cells?
  10. To make the observations mechanistically sound, authors should use the animal model and see if results especially in figure 5 remain consistent.
  11. Authors should include a hypothesis figure for better understanding.

Minor comments:

  1. Authors should note the catalog number of all used kits/reagents.
  2. Primer sequences used for cloning of sgRNAs should be noted.

Author Response

We upload response for review's comments.

Reviewer 2 Report

Peer Review: SMARCA4 Depletion Induces Cisplatin Resistance by  Activating YAP1-Mediated Epithelial-to-Mesenchymal Transition in Triple Negative Breast Cancer

In the manuscript authors have identified that SMARCA4 mutations to have a critical role in cisplatin resistance and metastasis in triple-negative breast cancer (TNBC) by inducing a mesenchymal-like subtype and activation of  Hippo-YAP/TAZ pathway. They also show by in vitro studies that YAP1 inhibitor verteporfin can be a novel therapeutic option for cisplatin-resistant patients. However the authors need to address a few major comments.

Comments:

  1. In fig 1B the authors compare the mutations in components of SWI/SNF complex between metastatic and primary TNBCs and show that ARID2A and SMARCA4 mutations are associated with metastasis. Then in Fig 2C they show the comparison of progression free survival of TNBC patients receiving cisplatin therapy and found that the survival is poor in the non-responder group. But this does not suggest that cisplatin resistance and poor patient outcome are due to the SWI/SNF mutations in the non-responder group. A correlation analysis between SMARCA4 mutations and survival analysis would be required to suggest this conclusion. Does SMARCA4 mutation also lead to reduced expression in TNBC patients? This can be analyzed using gene expression data from the patients included in the study or from TCGA?
  2. In figure it is unclear what the Y-axis denotes to depict SMARCA4 expression.
  3. In figure 3A, the authors have shown the presence of SMARCA4 sgrna sequence in the control sample too instead of the NT sgrna. This needs to be corrected.
  4. In figure 3C authors show that over expressing SMARCA4 in the KO cells can reduce the IC 50 to cisplatin treatment. Western blot results confirming expression of SMARCA4 in the KO cells need to added to Figure 3B.
  5. In Figure 5A, the P values of the comparison between control and SMARCA4 KO groups suggests no significant changes. Is this a typo error? If not the rationale for using YAP inhibitors in TNBCs of this subtype is not justified.
  6. An important drawback of this study is that there is no in vivo data that is presented to show the relevance of SMARCA4 KO in tumor growth. Also whether verteporfin inhibits tumor growth of SMARCA4 KO TNBCs in an in vivo tumor microenvironment needs to be studied. This is essential to propose YAP1 - targeted therapy for MSL-type TNBC harboring SMARCA4 downregulation.
  7. A recently published study (Mehta et al, npj Breast Cancer, 2021) have shown that SOX4 and SMARCA4 cooperatively regulate PI3K/Akt signaling and play an essential role in TNBC genesis and/or progression. But in the current study authors are showing that SMARCA4 deletion in TNBCs lead to more aggressive tumors. They need to discuss the differences between their study and the published study on SMARCA4 in TNBCs.
  8. Another report by Jose A. Guerrero-Martinez & Jose C. Reyes, Scientific reports, 2018 also showed that high expression of SMARCA4 was significantly associated with a poor prognosis in breast and ovarian cancer. This study also needs to be discussed based on the context if SMARCA4 inactivation in the MSL subtype.

Author Response

We upload response for reviewer's comment.

Round 2

Reviewer 1 Report

I congratulate the authors for the modifications. My concerns are partly answered. However, the authors have provided sufficient justifications. I just have a minor concern, drug resistance is the main concern of this manuscript and it is not emphasized in the hypothesis figure. I suggest authors ao add downstream components for a better understanding of the concept. 

Author Response

We appreciated reviewer’s detailed comment to improve our manuscript.

We also recognized your point, and modified hypothesis figure. Updated figure includes the content of cisplatin therapy and delineate its resistance overcome process by verteporfin therapy. We also revised its figure legend, slightly, but not highlighted to red font for editor’s convenience.

Round 3

Reviewer 1 Report

All my concerns are addressed. Now I recommend accepting this manuscript.